# Assessing the Link between Nurses’ Proficiency and Situational Awareness in Neonatal Care Practice Using an Eye Tracker: An Observational Study Using a Simulator

**DOI:** 10.3390/healthcare12020157

**Published:** 2024-01-09

**Authors:** Masahiro Sugimoto, Michiko Oyamada, Atsumi Tomita, Chiharu Inada, Mitsue Sato

**Affiliations:** 1Institute for Advanced Biosciences, Keio University, Tsuruoka 997-0052, Japan; 2Institute of Medical Sciences, Tokyo Medical University, Shinjuku, Tokyo 160-0022, Japan; tomita@tokyo-med.ac.jp; 3Faculty of Human Care Department, Tohto University, 1-1 Hinode-cho, Numazu 410-0032, Japan; michiko.oyamada@tohto.ac.jp; 4Department of Nursing, Nihon Institute of Medical Science, Iruma 350-0435, Japan; 5Faculty of Nursing, Japanese Red Cross College of Nursing, 4-1-3 Hiroo, Shibuya, Tokyo 150-0012, Japan; c-inada@redcross.ac.jp; 6Department of Nursing, Kiryu University, Midori 379-2392, Japan; sato-mi@o.kiryu-u.ac.jp

**Keywords:** nursing, eye tracking, neonatal care, midwifery

## Abstract

Nurses are expected to depend on a wide variety of visually available pieces of patient information to understand situations. Thus, we assumed a relationship between nurses’ skills and their gaze trajectories. An observational study using a simulator was conducted to analyze gaze during neonatal care practice using eye tracking. We defined the face, thorax, and abdomen of the neonate, the timer, and the pulse oximeter as areas of interest (AOIs). We compared the eye trajectories for respiration and heart rate assessment between 7 experienced and 13 novice nurses. There were no statistically significant differences in the time spent on each AOI for breathing or heart rate confirmation. However, in novice nurses, we observed a significantly higher number of instances of gazing at the thorax and abdomen. The deviation in the number of instances of gazing at the face was also significantly higher among novice nurses. These results indicate that experienced and novice nurses differ in their gaze movements during situational awareness. These objective and quantitative differences in gaze trajectories may help to establish new educational tools for less experienced nurses.

## 1. Introduction

Eye tracking is a unique tool for understanding situational awareness, decision-making, and behavior by analyzing eye gaze [1,2]. This tool could become a new educational method that complements conventional textbook-based education through feedback on gaze traits. In particular, this tool could potentially transfer experiential skills, such as complex decision-making and unconscious behavior based on visual information, from experienced to novice persons. In medicine, nursing, and other medical professions, many situations that need to be confirmed or recognized cannot be described in writing, and require extensive and diverse experience to acquire advanced skills.

Eye tracking is often used to analyze the gaze of skilled personnel during medical surgeries and diagnostic imaging [3,4]. Cognitive patterns used to elucidate clinical decision-making for evaluating patients’ deteriorating conditions have been assessed based on participants’ gaze traits [5]. A comprehensive assessment of the patient’s situation is a complex process; not only is the affected area determined through visual inspection, but so are the vital signs and surroundings. Thus, establishing a quantitative, objective method for analyzing gaze trajectory patterns and providing feedback to participants will lead to more efficient clinical training.

Two types of gaze analysis have been reported in the nursing field: (1) eye movement and pupil analysis and (2) gaze trajectory analysis. As an example of (1), the eye movements of intensive care unit nurses were measured to analyze the differences in their stress between day and night shifts. These eye movements were higher during the first handover phase of a nursing shift [6]. As an example of (2), operating room nurses were evaluated through gaze analysis, including their ability to work with various medical devices and avoid the risk of bacterial infection [7]. A comparison of gaze patterns during intravenous injection between skilled nurses and nursing students found that students spent relatively more time gazing at the needle and less time switching their attention [8]. Differences in gaze trial results were evaluated among nurses with different skill levels in the routine assessment of patients’ vital signs and the use of oxygen delivery equipment in simulator-based training [9]. When nurses obtained visual information from electronic health records, their gaze trajectory patterns were analyzed [10]. Differences were examined in terms of the degree to which nurses and physicians focused on the level of hand cleanliness and gaze in two scenarios related to intravenous injections [11]. The novel findings resulting from these experiments are attributed to the examination of multifaceted gaze trajectories. It is imperative to explore the appropriate analytical methods for each specific situation.

We also explored the differences in gaze between skilled nurses and students when determining the status of intravenous injections. Although the areas of interest (AOIs) were similar among the participating nurses, the gaze trails visualized in the network showed apparent differences, such as the fact that the students rarely checked vital signs despite the short overall nursing time, whereas experienced nurses frequently checked vital signs between each task [12]. Eye tracking-based gaze analysis has also been implemented in maternal and midwifery nursing. Visual attention during neonatal resuscitation care was investigated using eye tracking [13]. In an airway management scenario using a neonatal simulator, nurses with varying skill levels differed in the pattern of time spent on eye gaze dwell locations [14]. The application of eye-tracking analyses has been examined in nursing fields; however, additional experiments are required to assess the potential of this technology to understand the situational awareness of experienced nurses.

In maternity care, there are limited opportunities to gain timely hands-on experience in childbirth and neonatal care. Therefore, simulator-based practice with more efficient educational tools is required to overcome this problem. In the present study, we explored the differences between more and less skilled nurses with respect to assessing neonates’ status. We hypothesized that the trajectory of their gaze differs according to skill level. This experiment relates to basic maternal care skills, which require a high degree of situational awareness.

## 2. Materials and Methods

### 2.1. Study Design

This experiment was an observational study. Licensed nurses participated in this study, for which we used the Neonatal Vital Signs Simulator Model II LM-098 (Koken Co., Ltd., Tokyo, Japan), which reproduces the heartbeat and crying. We placed the simulator on a towel, with a diaper, timer, and pulse oximeter nearby (Figure 1a). The experiments were conducted in a laboratory at Tokyo Medical University. All nurses were recruited from the faculty at Kiryuu University, Toho University, and the Japanese Red Cross College of Nursing using convenience sampling. Benner’s theories, a common framework in nursing education, clarify skill acquisition and the role transition process undertaken by nurses from novice to expert [15]. Due to the transition of skill acquisition, 10 years of experience is required to reach the expert level [16,17]. Therefore, in this study, we deemed nurses with more than 10 years of midwifery experience experts (expert: E) and the others novices (novice: N). The study designer and data analyst were not involved as participants.

The nursing care scenario was as follows: A nurse was instructed to “perform a general observation for the first two hours of life to determine if the neonate can be transferred to the general ward for management”. The neonate progressed well; however, the neonate faced potential risks, and the nurses were required to evaluate whether these factors would manifest in the future. An observation procedure was not indicated. No restrictions on body or head movements were instructed.

### 2.2. Eye Tracking

For this study, we used ViewTracker3 (DITECT Co. Ltd., Tokyo, Japan) to monitor eye movements. This device has been used for various tasks in diverse fields; it has a front-facing camera and two side-facing cameras that capture eye movements (Figure 1b). Before the task, eye tracking was calibrated for each participant. They were instructed to maintain head movement and fixate on circles shown in different areas of the monitor connected to the eye-tracking device. Participants were later required to look at several points outside the monitor to confirm the consistency between actual and monitored gazes.

### 2.3. Data Collection

The configuration of ViewTracker3 was the same as that used in a prior study [12]. A front camera with a resolution of 1280 × 720 pixels and frame rate of 30 Hz was used. Two pupil cameras with a resolution of 192 × 192 pixels and frame rate of 120 Hz were used. Furthermore, the edge intensity (dimensionless) was set to 23, eye size (dimensionless) was measured as 10–150, and sensitivity for the detection of the black iris (dimensionless) was set to 0.997. The software ViewTracker3 (ver. 1.0520; DITECT, Tokyo, Japan) was employed with default configurations in high-resolution mode (M-JPEG). This software can identify eye-tracking data with an accuracy of one pixel. However, the actual accuracy depends on the distance between the object and the eye. The manual is available at https://www.ditect.co.jp/en/ (accessed on 14 December 2023).

The device was connected to a personal computer (PC) with the following configurations using a USB3 interface: operating system: Windows 64bit Pro; CPU: In-tel^®^Core™i7-8650U CPC@1.90 GHz 2.11 GHz; memory: 16.0 GB DDR4 2400 MHz; monitor: 12.5 inches (1920 × 1080 pixels); and storage: SSD: 512 GB M.2 2280 PCIe. We performed video analysis of the gaze measurement outcomes using View Tracker3 ver. No. 1.0520 (Detect Inc., Tokyo, Japan).

The gain area analysis function targeted an area to calculate the duration of gazing and the number of gaze entries from outside the area. The face, thorax and abdomen, timer, and monitor were designated as AOIs.

We obtained gaze trajectories using a gaze analysis function. Gaze and gaze movements were recorded. These settings were set as default. A minimum radius of 20 mm and a maximum radius of 60 mm for the circle was used. Staring within a range of 100 pixels of an AOI was considered to constitute gazing at the AOI. A different study used 150 ms as a threshold to define gazing at an AOI [18]. In this study, staring that took place for less than 100 ms was ignored. Gazing in off-screen areas was also ignored.

We compared quantitative data between the two groups using the Mann–Whitney U test. We employed the F-test to test for differences in variance between the two groups. We utilized GraphPad Prism software (ver. 9.5.1, San Diego, CA, USA) to conduct statistical analysis and visualize the data in box plots.

#### Ethical Considerations

We conducted this study in accordance with the Declaration of Helsinki; it was approved by the Nihon Institute of Medical Science (protocol code 20220001, 15 April 2022). We obtained written informed consent from each participant prior to their participation.

## 3. Results

### 3.1. Overview

Twenty licensed midwives participated in this study. Of these, we deemed 7 as E and the other 13 as N. Table 1 presents the characteristics of the participants.

### 3.2. Overall Care Time

Figure 2 displays examples of data recorded using ViewTracker (ver. 1.0530). Examples of the trajectory of the gaze data collected from experts (Figure 2a) and novice nurses (Figure 2b) are shown. The circled areas indicate the AOIs where the gaze remained for a certain duration, while the lines between circles imply movement of the gaze. The AOIs were not widely scattered over the entire area but tended to cluster in some fairly small areas. In Figure 2a, the AOIs are concentrated on the phonocardiograph, timer, and pulse oximeter, with the gaze moving back and forth between these areas. As outlined in Figure 2b, the AOIs tended to be concentrated on the face and chest.

Figure 2c,d present the outcomes of the area analysis. Figure 2c,d show sample data collected from expert and novice nurses, respectively. As most nurses were staring at four points—the face, thorax and abdomen, timer, and pulse oximeter—we designated these four areas as AOIs (green). We evaluated the duration for which the gaze was contained in these areas and the number of times the gaze entered these areas.

We compared the overall time required for nursing care between experts and novices. The median time was greater for experts, but the difference was not significant (*p* = 0.536; Mann–Whitney U test) (Figure 3a). The time required to assess breathing revealed no significant difference (*p* = 0.757) (Figure 3b), and the time required to evaluate the heartbeat also showed no significant difference (*p* = 0.699) (Figure 3c). The difference in variance determined using the F-test demonstrated a significant difference in overall time (*p* = 0.017) (Figure 3a). In general, novices tend to have less variance in time, whereas experts tend to have more variance in time.

### 3.3. Duration of Gaze Entering Each Area

Figure 4 presents the outcomes of the comparison between experts and novices with respect to total gaze time in each area. Figure 4a–h portray the data obtained during the evaluation of breathing and heartbeat, respectively. Figure 4a,e show data for the face; Figure 4b,f show data for the thorax and abdomen; Figure 4c,g show data for the timer; and Figure 4d,h present data for the pulse oximeter.

The most common findings for the respiration and heart rate evaluation were that most of the time was spent on the thorax and abdomen (Figure 4b,f), followed by the face (Figure 4a,b). Timers (Figure 4c,g) and pulse oximeters (Figure 4d,h) required less gazing. In the heartbeat evaluation, experts spent a longer time gazing at timers and pulse oximeters (Figure 4c,d,g,h). In any case, there was no statistically significant difference in the time between the experts and novices.

We examined differences in variability using the F-test, which was statistically significant at *p* = 0.0200 for the pulse oximeter (used for breath confirmation). These results indicate that variability was significantly greater for novices than for experts.

### 3.4. Number of Times the Gaze Entered Each Area

Figure 5 compares the number of times the gaze entered each area. Figure 5a–d show respiration data, and Figure 5e–h display the heartbeat evaluation. Figure 5a,e present the face data. Figure 5b,f depict the thorax and abdomen data, Figure 5c,g show the timer data, and Figure 5d,h portray the pulse oximeter data. The respiration and heartbeat assessments showed more entries into the gaze area than into the other areas (Figure 5b,f).

There was no statistically significant difference in the respiration data between the experts and novices (Figure 5a–d). The heartbeat assessment data revealed that the number of novice nurses was significantly higher than that of experts (*p* = 0.0348, Mann–Whitney U test, Figure 5f).

The differences in variability were statistically significant (*p* = 0.0042, F-test) for the pulse oximeter area in the respiratory evaluation (Figure 5d). Variability was significantly greater for experts than for novices. However, for breath assessment, we noted a significant difference in variability in the facial area (*p* = 0.00432). The variability was significantly greater for novices than for experts.

## 4. Discussion

We used eye-tracking techniques to analyze the differences in gaze trajectories between skilled and novice nurses in neonatal nursing care. There were no significant differences between the two groups with respect to the overall time spent on nursing care or time spent on each AOI. However, the variance in the overall time spent on nursing care was significantly greater for experts. It is possible that the novices had less variance in their nursing care methods, while the experts’ techniques may have been individually optimized based on their long-term nursing experience.

Although there were no significant differences in respiratory confirmation between the areas, the number of glances entering each area was significantly greater for novices than for experts in the thoracic and abdominal areas. Likewise, the variance in the number of times a gaze entered the face area during breath confirmation varied more among novices.

In a gaze analysis of nurses and nursing students performing intravenous injections, nurses spent more time looking at the needle but less time looking at other individual areas and comprehended the situation better based on visual information in wider areas in a shorter period than nursing students [19]. In our study on nursing care for intravenous drip confirmation, skilled nurses moved their eyes broadly around the patient’s face and arms, the intravenous drip-related equipment, and other areas in their surroundings to obtain a holistic view of the situation compared to nursing students and new nurses [12]. Comparisons of nursing skills between experienced and novice nurses resulted in a shorter gaze duration for several AOIs in the prepared scenarios [20,21]. In the present study, the number of times that experts looked at the thoracic and abdominal areas during respiratory confirmation was significantly lower than that of the novices. This implies that skilled nurses may not only focus on the thoracic and abdominal areas when providing nursing care; they may also move their gaze over a wide range of areas to understand the entire situation.

Situational awareness in nurses is essential for timely clinical decision-making. Many studies have used gaze analysis to quantitatively evaluate these factors. No statistically significant differences in gaze dwell time or gazing time between skilled and novice nurses have been reported in terms of the AOI of patients in a simulated nursing environment [9,22]. In a simulator-based neonatal resuscitation scenario, most of the gaze time was focused on the equipment, followed by the neonate, with the least amount of time spent on the monitor. In this study, more time was spent on the thorax and abdomen during respiratory and heartbeat assessments, followed by the face and newborns’ vital signs. The face tended to be looked at more often than the timer regarding the number of times the eyes examined these AOIs during respiratory confirmation, but the opposite was true for heartbeat assessments. Thus, we can infer that the newcomers moved their gaze not only to the thorax and abdomen but also to other AOIs more frequently in the heartbeat assessments, and that they attempted to obtain a broader range of information. It is possible that skilled nurses place more emphasis on the thoraco-abdominal region or consider information such as palpation in confirming the heartbeat. However, eye-tracking analysis alone cannot confirm this hypothesis.

Several limitations should be acknowledged. Although the use of simulators in skills practice related to postpartum neonatal nursing has shown changes in communication and confidence before and after practice [23], we only analyzed the differences between expert and novice nurses. An observational study comparing expert and novice nurses is not sufficient to evaluate the educational effect of using the eye tracker. These two groups have different co-factors, although age did not show a significant impact in this study. Comparison with/without eye-tracking-based training is required. A recent study examined differences in gaze within a nursing team during a precardiac arrest simulation and reported that designated nurses spent more time checking the status of signs of patient deterioration than other nurses [24]. We could not target such role differences and communication within organizations among multiple nurses. In addition, although a simulator was used to reproduce heartbeats, the simulator and scenario did not allow for unpredictable events. We also did not analyze information obtained from voice and tactile sensations. Although eye movement partially reflects the cognitive process in the diagnostic problem of medical imaging with gaze analysis, some limitations have been reported in which gaze tracking data cannot directly assess cognitive processes [25]. In this study, we did not limit the body and head movements of each participant. The participants’ bodies and heads remained still during the assessment of breathing and heartbeat. However, the head moved in other processes, such as checking the condition of the toes of neonates. The current analysis only works well for small movements of the head and body. Due to the lack of a preliminary study, the sample size of this study was not estimated theoretically. The other problem is convenient sampling, which is the recruitment method used in this study. A validation study should be conducted with a random sampling of participants in order to generalize the results. In the future, it will be necessary to develop a system that allows for an integrated analysis of information other than gaze. The eye-tracking study used a visual patient avatar to analyze the interpretation of vital signs and situational awareness of physicians and nurses [26]. Such a digital tool, complementary to the simulator used in this study, would help generate various dynamic scenarios. This study designed comparison groups based only on midwifery experience. Variable bias is still present, and more participants should be included to analyze the relationship between co-factors and observed outcomes. External validation with additional participants across institutions and scenarios is needed for more generalizable results.

There remains a gap between nursing performance using a simulator and actual care in clinical settings, which implies that there is a requirement to improve simulation-based practice [27]. Therefore, using eye tracking in a clinical setting is considered one of the ways to address this problem. However, in such a situation, about half of the data in the scenario are outside of the monitorable range due to intense eye movement [18]. Thus, the situation and scenario must be carefully designed while considering the characteristics of the gaze analysis device.

## 5. Conclusions

In this study, we hypothesized that experienced and novice neonatal nurses would differ in their gaze trajectories. Thus, we explored the differences in gaze between nurses with different skill levels in the nursing care of neonates. We set the neonate’s face, thorax and abdomen, timer, and pulse oximeter as the AOIs, and compared the time spent gazing at these and the number of times the participants’ gaze started to enter each AOI. There was no statistically significant difference in the time spent on each AOI for either respiratory or heart rate confirmation. However, there was a statistically significant difference in the number of times a gaze was focused on the thoraco-abdominal AOI for heart rate confirmation. These results show differences between highly skilled and new nurses during status checks. These findings cannot be obtained simply based on the time spent on each AOI. In the future, it will be necessary to establish an objective interpretation method for gaze analysis as an educational tool for less skilled nurses. Understanding the difference between expert and novice gaze trajectories can help the novice understand the situational expertise that the expert possesses.

## Figures and Tables

**Figure 1 healthcare-12-00157-f001:**
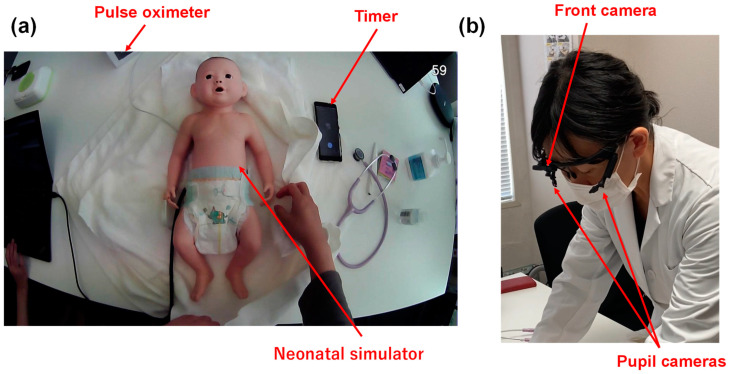
Nursing care setting. (**a**) The neonatal simulator is set up on a towel and in a diaper. A timer, pulse oximeter (mock-up), and phonocardiograph are placed nearby. (**b**) Eye-monitoring device. One camera in the front and two pupil-taking cameras on the side monitor the trajectory of the gaze.

**Figure 2 healthcare-12-00157-f002:**
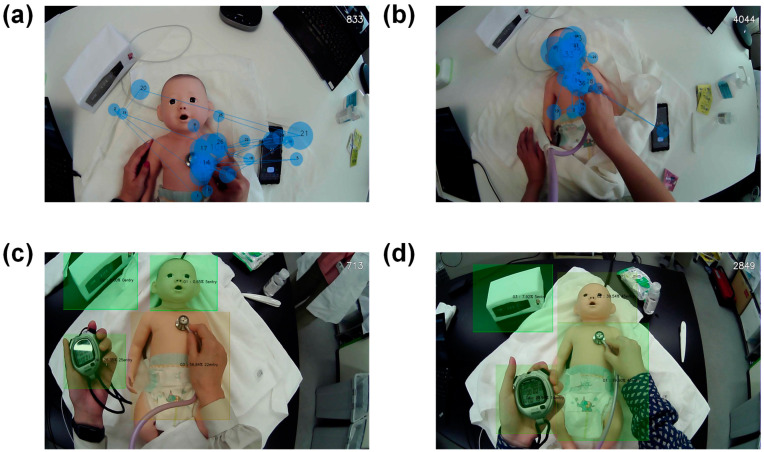
Examples of analysis of logged gaze trajectories. (**a**) Gaze trajectory of an expert nurse. The blue circles indicate gazes (fixations), their size indicates their duration, and the numbers inside these circles determine their order of occurrence. (**b**) Novice nurse’s gaze trajectory. (**c**) An example of an expert nurse’s area-based analysis. The time that the gaze is in the area highlighted in green and the number of times the gaze enters this area was counted. (**d**) Novice nurse’s area-based analysis.

**Figure 3 healthcare-12-00157-f003:**
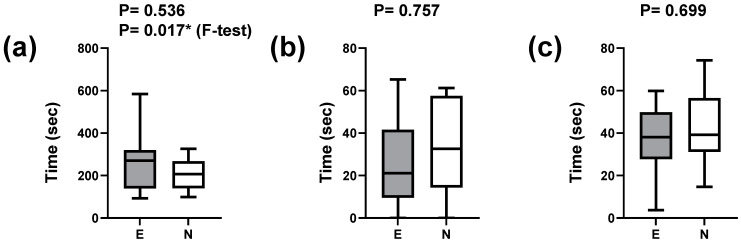
Comparison of time spent on neonatal nursing care. (**a**) Overall care time. (**b**) Time to assess respiration. (**c**) Time to assess the heartbeat. The *p*-values in the first row are the results of the Mann–Whitney U test. The *p*-values in the second row are the results of the F-test and are shown only when *p* < 0.05. Horizontal lines in box plots indicate quartiles as well as maximum and minimum values. * *p* < 0.05.

**Figure 4 healthcare-12-00157-f004:**
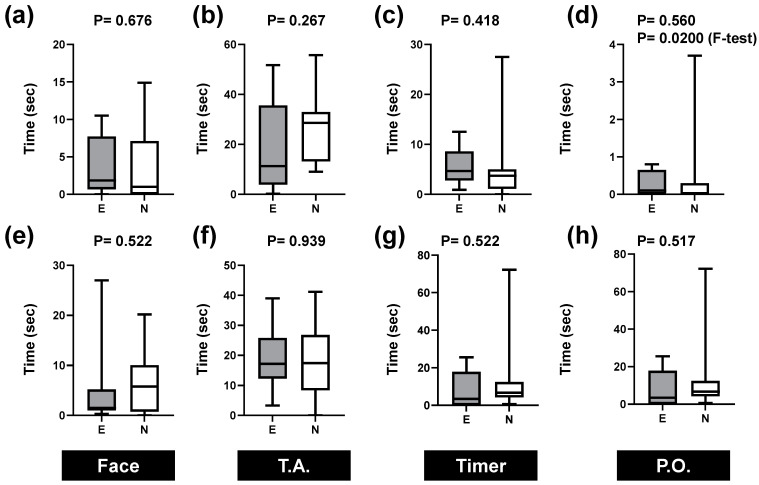
Comparison of gazing time in the specified areas during neonatal nursing care. Times for evaluation of (**a**–**d**) breathing, (**e**–**h**) heartbeat, (**a**,**e**) face, (**b**,**f**) thorax and abdomen (T.A.), (**c**,**g**) timer (**d**,**h**), and pulse oximeter (P.O.). The *p*-values in the first row are the results of the Mann–Whitney U test. The *p*-values in the second row display the outcomes of the F-test and are shown only when *p* < 0.05. The horizontal box lines in plots indicate quartiles and maximum and minimum values.

**Figure 5 healthcare-12-00157-f005:**
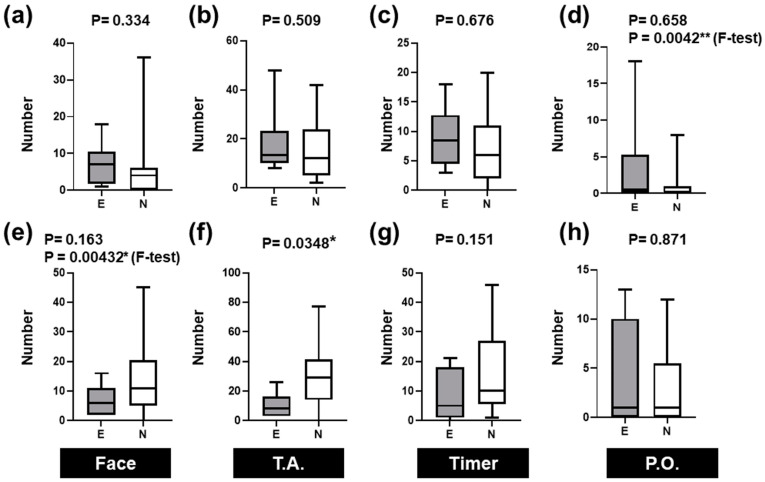
Comparison of the number of times the gaze entered the specified areas during neonatal nursing care. (**a**–**d**) Times for evaluation of breathing, (**e**–**h**) heartbeat, (**a**,**e**) face, (**b**,**f**) thorax and abdomen (T.A.), (**c**,**g**) timer, and (**d**,**h**) pulse oximeter (P.O.). The *p*-values in the first row are the outcomes of the Mann–Whitney U test. The *p*-values in the second row are the results of the F-test and are shown only when *p* < 0.05. The horizontal box lines in the plots indicate the quartiles and maximum and minimum values. * *p* < 0.05, ** *p* < 0.01.

**Table 1 healthcare-12-00157-t001:** Characteristics of the participants.

	Expert ^1^	Novice ^1^	*p*-Value ^2^
Experience			
Nurses	23.1 ± 9.41	9.84 ± 8.95	0.00135
Midwifery nurses	22.7 ± 2.75	3.69 ± 3.98	<0.0001
Sex			
Female/male	7/0	13/0	-
Ages			
60s	1	0	0.138
50s	2	1	
40s	3	3	
30s	1	4	
20s	0	5	

^1^ Years (average ± standard deviation). ^2^ Mann–Whitney U test for experience and χ^2^ test for ages.

## Data Availability

The data presented in this study are available upon request from the corresponding author.

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
