# Peer review of "Assessing the Link between Nurses’ Proficiency and Situational Awareness in Neonatal Care Practice Using an Eye Tracker: An Observational Study Using a Simulator"

_healthcare, 2024, doi:10.3390/healthcare12020157_

Round 1
Reviewer 1 Report (Previous Reviewer 1)
Comments and Suggestions for Authors
In the article "The link between nurses’ proficiency and situational awareness in neonatal care practice using an eye tracker: An observational study using a simulator", the authors explored the differences in gaze trajectories for respiration and heart rate assessment between two groups of nurses with a different level of professional experience: more and less experienced ones. The researchers conducted an experiment in which they take into account three metrics i.e. the time spent on neonatal nursing care, the dwell time and the number of times of the gaze entering the specified AOI. The authors have shown statistically the difference between experts and novices in the number of times both groups looked at the thoraco-abdominal AOIs during respiratory confirmation. Moreover, the authors indicate the difference in the number of times gazing toward the face of the neonatal simulator among both researched groups.
The topics discussed in the article are interesting and useful as objects of research. The results, statistical analysis, discussion and conclusions were properly and accurately described. Summing up, the paper is professionally written and therefore I recommend a few minor corrections following the comments below.
· I suggest writing the phrase "eye tracking" separately without a hyphen while phrases where "eye tracking" functions as an adjectival modifier, i.e. when it precedes the noun it modifies, e.g. in "eye-tracking data" "eye tracking" should be written with a hyphen. This issue should be unified throughout the document.
· L165-167: The style at the beginning of chapter 3.2 needs correcting by eliminating the repetition of "Figure 2ab".
· L239: The font size of the numerical and text labels in Figure 5h should be increased.
· The references are incomplete, doi is missing in many points.
Author Response
In the article "The link between nurses’ proficiency and situational awareness in neonatal care practice using an eye tracker: An observational study using a simulator", the authors explored the differences in gaze trajectories for respiration and heart rate assessment between two groups of nurses with a different level of professional experience: more and less experienced ones. The researchers conducted an experiment in which they take into account three metrics i.e. the time spent on neonatal nursing care, the dwell time and the number of times of the gaze entering the specified AOI. The authors have shown statistically the difference between experts and novices in the number of times both groups looked at the thoraco-abdominal AOIs during respiratory confirmation. Moreover, the authors indicate the difference in the number of times gazing toward the face of the neonatal simulator among both researched groups.
The topics discussed in the article are interesting and useful as objects of research. The results, statistical analysis, discussion and conclusions were properly and accurately described. Summing up, the paper is professionally written and therefore I recommend a few minor corrections following the comments below.
Thank you for your valuable time and insights into improving our manuscript. We are truly grateful.
- I suggest writing the phrase "eye tracking" separately without a hyphen while phrases where "eye tracking" functions as an adjectival modifier, i.e. when it precedes the noun it modifies, e.g. in "eye-tracking data" "eye tracking" should be written with a hyphen. This issue should be unified throughout the document.
We thank this observation. The manuscript was revised throughout accordingly (L 28, 105, 109, ).
- L165-167: The style at the beginning of chapter 3.2 needs correcting by eliminating the repetition of "Figure 2ab".
According to these comments, we revised the sentence (L162-163).
- L239: The font size of the numerical and text labels in Figure 5h should be increased.
As pointed out, the font size of Figure 5h may appear smaller than the others. In this manuscript, we have inserted a PNG image instead of the PowerPoint clipboard to solve this problem (Figure 5).
- The references are incomplete, doi is missing in many points.
We added doi to the references 18, 29, 21, and 27.
Reviewer 2 Report (Previous Reviewer 3)
Comments and Suggestions for Authors
The authors revised their manuscript based on the reviewer's comments and suggestions, which increased the scientific value as well as the readability of the manuscript. Thank you for your patience. The reviewer appreciates the effort made by authors. The revision is satisfactory, and this reviewer recommend this manuscript to be published in Healthcare, pending the resolution of the several issues raised below:
1.
The authors did not address comment #13 and #14 sufficiently. As for the comment #13, the corresponding parts does not appear in the revised manuscript. Basically, the author's responses should be incorporated into the revised manuscript. The reviewer understands that the authors were unable to estimate the precise sample size because of lack of previous study. However, this fact should be acknowledged in the limitation section. In addition, this study may be used for sample size estimation for the further clinical study.
Turning to comment #14, please see the lines 307-312 of the revised manuscript. How missing data were addressed was not shown in these lines. Please elaborate. The article by Viautour et al (PMID: 37998568) may be helpful when addressing this comment.
2.
Limitation section
If the same researcher was involved in participant enrollment, outcome measurement, and statistical analysis, the authors should acknowledge that there is a theoretical risk of biased assessment. Please provide the information who measured the main outcomes shown on Figure 3 to 5, performed and the statistical analysis.
3.
Unfortunately, the authors did not discuss the generalizability (external validity) of the study results despite the reviewer's comment in the previous review round. Please discuss this point. In this reviewer’s opinion, the generalizability of the study findings are quite limited, because of the convenient sample, observational study at a single institution, etc. Please discuss greater in details.
4.
Recruitment
The bias that was introduced by the convenient sampling strategy should be discussed greater in details.
Author Response
The authors revised their manuscript based on the reviewer's comments and suggestions, which increased the scientific value as well as the readability of the manuscript. Thank you for your patience. The reviewer appreciates the effort made by authors. The revision is satisfactory, and this reviewer recommend this manuscript to be published in Healthcare, pending the resolution of the several issues raised below:
Thank you again for your valuable time and insights into improving our manuscript. We are truly grateful.
1.
The authors did not address comment #13 and #14 sufficiently. As for the comment #13, the corresponding parts does not appear in the revised manuscript. Basically, the author's responses should be incorporated into the revised manuscript. The reviewer understands that the authors were unable to estimate the precise sample size because of lack of previous study. However, this fact should be acknowledged in the limitation section. In addition, this study may be used for sample size estimation for the further clinical study.
According to your comments, we added these issues to the limitation (L303-304).
Turning to comment #14, please see the lines 307-312 of the revised manuscript. How missing data were addressed was not shown in these lines. Please elaborate. The article by Viautour et al (PMID: 37998568) may be helpful when addressing this comment.
We appreciate this comment. We revised these sentences by citing the reference (L308-311).
2.
Limitation section
If the same researcher was involved in participant enrollment, outcome measurement, and statistical analysis, the authors should acknowledge that there is a theoretical risk of biased assessment. Please provide the information who measured the main outcomes shown on Figure 3 to 5, performed and the statistical analysis.
We revised the material and method section (L97-98) to describe that “The study designer and data analyst were not involved as participants.”
3.
Unfortunately, the authors did not discuss the generalizability (external validity) of the study results despite the reviewer's comment in the previous review round. Please discuss this point. In this reviewer’s opinion, the generalizability of the study findings are quite limited, because of the convenient sample, observational study at a single institution, etc. Please discuss greater in details.
According to your comments, we added these issues to the limitation (L311-315).
4.
Recruitment
The bias that was introduced by the convenient sampling strategy should be discussed greater in details.
According to your comments, we added these issues to the limitation (L305-307).
This manuscript is a resubmission of an earlier submission. The following is a list of the peer review reports and author responses from that submission.
Round 1
Reviewer 1 Report
Comments and Suggestions for Authors
In the article "The link between nurses’ proficiency and situational awareness in neonatal care practice using an eye tracker", the authors explored the differences in gaze between nurses with different skill levels in the nursing care of neonates. The researchers conducted an experiment in which they take into account three eye-tracking metrics: time spent on neonatal nursing care, dwell time and the number of times of the gaze entering the specified AOI. The researchers have shown statistically the difference in the number of times a gaze was moved to the thoraco-abdominal AOI for heart rate confirmation.
The topics discussed in the article are interesting and useful as objects of research. The results, statistical analysis, discussion and conclusions were properly and accurately described. Summing up, the paper is well written and therefore I recommend the submitted manuscript for publication. However, the authors should make a minor revision following the comments below.
General comments:
· Adding a subchapter related to metrics "Overall care time" in line 152 and labelling subchapter "Time of gaze entering each area" as 3.3 should be considered.
· Information on how the eye tracker was calibrated needs to be added.
· It is advisable to provide information on whether participants taking part in the study were instructed to stand still and not to move their heads. If not, the information on to what extent head or body movements were allowed in the experiment should be added.
· It would be advisable to specify the accuracy of the eye tracker used in the study if such information is available.
· It would be worth mentioning that in the future, similar studies on a larger number of participants, and if possible under real (not simulation) conditions should be conducted. In the analyses it might be interesting to use eye-tracking indicators other than time and the number of AOI entries.
Specific comments:
L92-94: If there are default settings for the device, it would be definitely advisable to provide the source for the information in this paragraph (manufacturer's website, documentation).
L111-114: This fragment is not comprehensible and needs to be revised.
L117: The information for what purpose GraphPad Prism was used needs to be provided.
L132-133, L141-145: The description of Figure 2ab should be revised, as the blue circles indicate gazes (fixations), their size indicates their duration, while the numbers inside these circles determine their order of occurrence. AOIs, on the other hand, are defined at the stage of the analysis, as shown in Figure 2cd.
L157: The P value given in the text should be corrected, as it is different in Figure 3a.
L175: In the text, the correctness of the labelling of the figures needs to be verified.
L197-202: When describing the results, the corresponding figure numbers should be given, as it is done in lines 152-159.
L205: The font size of the numerical and text labels in Figure 5h should be increased.
Comments on the Quality of English LanguageIn the article "The link between nurses’ proficiency and situational awareness in neonatal care practice using an eye tracker", the authors explored the differences in gaze between nurses with different skill levels in the nursing care of neonates. The researchers conducted an experiment in which they take into account three eye-tracking metrics: time spent on neonatal nursing care, dwell time and the number of times of the gaze entering the specified AOI. The researchers have shown statistically the difference in the number of times a gaze was moved to the thoraco-abdominal AOI for heart rate confirmation.
The topics discussed in the article are interesting and useful as objects of research. The results, statistical analysis, discussion and conclusions were properly and accurately described. Summing up, the paper is well written and therefore I recommend the submitted manuscript for publication. However, the authors should make a minor revision following the comments below.
General comments:
· Adding a subchapter related to metrics "Overall care time" in line 152 and labelling subchapter "Time of gaze entering each area" as 3.3 should be considered.
· Information on how the eye tracker was calibrated needs to be added.
· It is advisable to provide information on whether participants taking part in the study were instructed to stand still and not to move their heads. If not, the information on to what extent head or body movements were allowed in the experiment should be added.
· It would be advisable to specify the accuracy of the eye tracker used in the study if such information is available.
· It would be worth mentioning that in the future, similar studies on a larger number of participants, and if possible under real (not simulation) conditions should be conducted. In the analyses it might be interesting to use eye-tracking indicators other than time and the number of AOI entries.
Specific comments:
L92-94: If there are default settings for the device, it would be definitely advisable to provide the source for the information in this paragraph (manufacturer's website, documentation).
L111-114: This fragment is not comprehensible and needs to be revised.
L117: The information for what purpose GraphPad Prism was used needs to be provided.
L132-133, L141-145: The description of Figure 2ab should be revised, as the blue circles indicate gazes (fixations), their size indicates their duration, while the numbers inside these circles determine their order of occurrence. AOIs, on the other hand, are defined at the stage of the analysis, as shown in Figure 2cd.
L157: The P value given in the text should be corrected, as it is different in Figure 3a.
L175: In the text, the correctness of the labelling of the figures needs to be verified.
L197-202: When describing the results, the corresponding figure numbers should be given, as it is done in lines 152-159.
L205: The font size of the numerical and text labels in Figure 5h should be increased.
Reviewer 2 Report
Comments and Suggestions for Authors
Thanks for the opportunity to review the article " The link between nurses’ proficiency and situational awareness in neonatal care practice using an eye tracker".
Very interesting topic in the area of simulation, congratulations on your study. I make a few suggestions for changes that will benefit the study.
Abstract
This abstract fails to mention the type of study carried out.
They must not repeat key words with words that are already in the text.
Introduction
The aim of the study is not clear, can you please reformulate it?
Materials and methods
In this section there is still no information on the type of study! Could you please mention the type of study?
Results
They must write Figure 1 before the 2 figures appear
They must write Figure 2 before the 4 figures appear
They must write Figure 3 before the 3 figures appear
They must write Figure 4 before the 8 figures appear
They must write Figure 5 before the 8 figures appear
Discussion
I suggest you explore the differences between experts and novice nurses further in the discussion.
Years of experience should be discussed, why 7 or 10 years of experience? What does this mean?
This could be an important bias in analysing the data.
Conclusions
As the aim of the study is unclear, it is not clear from the conclusion whether it responds to the initial objective.
Comments on the Quality of English LanguageMinor editing of English language required
Reviewer 3 Report
Comments and Suggestions for Authors
I reviewed the article entitled “The link between nurses’ proficiency and situational awareness in neonatal care practice using an eye tracker” by Sugimoto et al. submitted to healthcare. In this simulation study, the authors mainly compared the eye trajectories for respiration and heart rate assessment between experienced and novice nurses. They found that the time spent on each areas of interests (AOIs) for breathing or heart rate confirmation were similar between these two groups, while times gazing at the thorax and abdomen, as well as the face was higher among novice nurses. From these observations, they claimed that quantitative differences in gaze trajectories may help to establish new educational tools for less experienced nurses. First, the reviewer pays respect for the Authors' tremendous effort spent on this manuscript. However, there are numerous concerns with the data presentation, design as well as the methodology. My concerns are listed below:
1
This observational study does not follow STROBE Statement Guidelines (https://www.equator-network.org/reporting-guidelines/strobe/). The authors should respect the basic rule of scientific writing.
2
Please indicate the study’s design with a commonly used term in the title, according to the STROBE check list. The title also should be more specific. One example is therefore “The link between nurses’ proficiency and situational awareness in neonatal care practice using an eye trackers: an observational study using a simulator"
Introduction
3
What is unknown and what is the knowledge gaps? It is difficult to follow. The introduction should also clearly indicate why the authors collected the data, the motivation of this studies, or how the information collected can be used to solve the current problems in neonatal nursing care.
4
At the end of the introduction section, please state the any prespecified hypotheses according to the STROBE check list.
Methods
5
The main exposure of this study is expert and novice nurses. This study deemed nurses with more than 10 years of midwifery experience as experts while the others as novices.
Please indicate the rationale of the definitions.
6
The main outcome of this study seems to be the spent times for the pulse oximeter, heartbeat evaluation, face, thorax, and abdomen. The most fundamental methodologic issue, which should be obvious to someone familiar with neonatal nursing care, is the lack of clinical meaning of measured outcome. There were no clear explanation regarding whether the observed differences were really clinical meaningful difference or not. Whether the measured outcomes were validated outcomes were also unclear. Since they are the main outcomes of this study, they authors should explain grater in details.
7
Present key elements of study design early in the method section, according to the STROBE Statement Guidelines
8
Clearly define all outcomes, exposures, predictors, potential confounders, and effect modifiers.
9
Describe all statistical methods, including those used to control for confounding.
10
Describe any efforts to address potential sources of bias, according to the STROBE checklist. For example, blinding is one of the attractive methods to reduce above mentioned biased assessment. If done, please provide who was blinded and how.
11
Please describe settings and locations more in details (e.g. a tertiary hospital, academic hospital, etc.) where the data were collected. This information should help readers to depict the context of this study more accurately. At the current form, it is difficult to image the study setting.
12
Who planned this study, who collected data, and who conducted the statistical analysis? I think if the same researchers are involved in study planning, data collecting, outcome measurement, and statistical analysis, there is a theoretical risk of biased assessment.
13
Sample size calculation is missing. Please explain how the study size was arrived at.
14
Explain how missing data were addressed. The recent study employing the similar methodology acknowledges that there are considerable numbers of the missing data [1].
15
How to recruit the experienced nurse and novice nurses? Convenient sampling?
16
The Participants flow diagram is missing. Please give exact numbers of participant potentially eligible, examined for eligibility, confirmed eligible, included in the study, completing follow-up, and analyzed.
17
Table 1. Baseline characteristics of the participants
Many vital information is missing. Give characteristics of study participants such as age, sex, board certification, etc. At this current form, many readers including myself find it difficult to image the characteristics of study subjects. In addition, these variables would have confounded the results. Why the authors have not attempted to adjusting for such confounders? There are too many unmeasured confounders in this study, which greatly hider the meaningful comparison. At the current form, this reviewer therefore cannot trust the central claim of your manuscript. The authors should adjust for such important confounders to provide more reliable data.
Discussion
18
The limitation section is missing. Please discuss limitations of the study, taking into account sources of potential bias or imprecision. Consider the important limitations and do not just list them but consider their relevance and how they might bias the results. Discuss both direction and magnitude of any potential bias.
19
What is the strength of this simulation-based study? Please indicate after the limitation section.
20
Please discuss the generalizability (external validity) and implications for practice of the study results.
Conclusion
21
The authors stated that "These results show differences between highly skilled and new nurses during status checks" in the conclusion section. How to use this conclusion to improve the clinical outcome of the patients?
The authors concluded that "it will be necessary to establish an interpretation method for gaze analysis as an educational tool for less skilled nurses." The, what is the meaning of this study?
Minor points
22
Keep abbreviations to a minimum. Do not use non-standard abbreviations unless they appear at least three times in the text.
Although the number of criticisms listed above, this reviewer should however state that it is laudable that this work is derived from huge efforts made by the authors, who are working as the frontline healthcare professionals. The reviewer respects the authors’ time and effort spent on this manuscript, and the authors ‘patience and professionalism in dealing with my comments.
1. Viautour J, Naegeli L, Braun J, Bergauer L, Roche TR, Tscholl DW, Akbas S. The Visual Patient Avatar ICU Facilitates Information Transfer of Written Information by Visualization: A Multicenter Comparative Eye-Tracking Study. Diagnostics (Basel). 2023 Nov 12;13(22):3432. doi: 10.3390/diagnostics13223432. PMID: 37998568; PMCID: PMC10670428.